# Anticipated Transport Choices in a World Featuring Autonomous Transport Options

Leon Booth [1,*], Victoria Farrar [1], Jason Thompson [2], Rajith Vidanaarachchi [2], Branislava Godic [2], Julie Brown [1], Charles Karl [3] and Simone Pettigrew [1]

1   The George Institute for Global Health, University of New South Wales, Sydney, NSW 2052, Australia
2   Melbourne School of Design, The University of Melbourne, Melbourne, VIC 3010, Australia
3   Australian Road Research Board (ARRB), Port Melbourne, Melbourne, VIC 3207, Australia
*   Correspondence: lbooth@georgeinstitute.org.au

**Abstract:** (1) Background: The automation of transport systems is well underway; however, it is unclear how this will affect people's mobility choices. Changes in these choices have implications for health and the sustainability and efficiency of transport systems, making it important to understand how the advent of autonomous vehicles might affect people's transport behaviors. The aim of the present study was to address this knowledge gap in the Australian context. (2) Methods: Respondents reported their demographic information, current transport behaviors, the perceived importance of transport-related factors, and attitudes toward autonomous vehicles. They then read a vignette describing a future scenario involving autonomous vehicles that was informed by expert stakeholders. After reading the vignette, the respondents selected those transport options that they would anticipate using in the depicted scenario. Descriptive analyses were conducted to examine changes in transport choices, while regression models were employed to identify the predictors of choices in the future scenario. (3) Results: Most respondents envisaged making greater use of active, shared, and public transport options in an autonomous future, compared to their current use of these options. The intended use of private transport options halved. The most consistent predictor for selecting a certain mode of transport was the current use of that option or its non-autonomous equivalent. (4) Conclusion: Overall, favorable changes in the envisaged use of transport were observed for the hypothetical scenario, which was characterized by improved public transport, a practical active transport infrastructure, and relatively cheap shared autonomous vehicles. If policymakers can act to realize these outcomes, the autonomation of transport is likely to lead to positive societal change.

**Keywords:** autonomous vehicles; transport choices; sustainability; health; physical activity; active transport; shared autonomous vehicles; private autonomous vehicles; public transport

## 1. Introduction

The automation of transport is well underway: autonomous passenger trains are already operational [1], autonomous freight ships are being tested [2], autonomous vehicle (AV) trials are in progress [3], and flying autonomous taxis are being developed [4]. AVs are predicted to comprise a substantial proportion of passenger vehicles by 2050 [5]. This paper will describe how AVs are expected to affect transport choices, outline a new approach for exploring this phenomenon, and report the resulting findings, along with their implications for the relevant policy decisions.

AVs have the potential to provide several advantages over conventional cars, including reduced traffic collisions, lower emissions via increased efficiencies in transportation networks, increased mobility for people who are unable to drive, and enhanced safety for vulnerable road users [6–8]. There are also potential disadvantages—AVs could reduce participation in active modes of transport, such as walking and cycling, increase

congestion due to empty vehicles traveling on roads, result in job losses, and foster urban sprawl [9–12].

The actual benefits and drawbacks of AVs are likely to depend on the AV options offered by the market and the choices that people make [9,10,13,14]. For example, it is anticipated that there will be two primary forms of AV passenger cars: (i) privately owned AVs that will be analogous to privately owned conventional cars, where the owner has exclusive use of the vehicle, and (ii) shared AVs that operate within taxi/ride-hail systems that people utilize as "mobility as a service" options, in preference to owning vehicles [15,16]. Compared to private AVs, it is typically predicted that shared AVs will yield more benefits, such as being more environmentally sustainable, minimizing congestion, and making transport more equitable through reduced costs [17,18].

Compared to both private and shared AVs, active modes of transport (e.g., walking and cycling) and public transport options are expected to result in fewer emissions per person in terms of distances traveled [10,19–21]. Active modes of transport have the added benefit of requiring physical activity, which can help to reduce incidence rates of non-communicable diseases and improve mental health outcomes [22–24]. Therefore, to maximize the benefits of AVs, it will be important to ensure that they do not undermine the use of active and public transport options [25].

Due to the uncertainties involved, it is difficult to predict transport modal shifts that may occur once AVs become widely available [3,11]. This has been reflected in modeling studies using different assumptions about autonomous futures [9]. Some have assumed that AVs will be cheaper to use than conventional cars, have shorter travel times than other options, decrease the value of travel time, and reduce waiting times for users. These studies typically predict that private and shared AVs will reduce the use of public transport and active transport [9,26,27]. In contrast, models assuming that private AVs will be prohibited, and that shared AVs will have longer waiting times during periods of peak demand, have predicted greater use of public transport and active transport options compared to current usage rates [27,28].

Another method that has been used to predict future transport choices concerns multi-stakeholder foresight approaches that involve experts being interviewed about their predictions regarding autonomous futures [3]. Stakeholders typically predict that if private AVs are available, they will be a popular option and will result in more vehicle miles being traveled [3,29,30]. Public transport use may increase if shared AVs complement these systems and act as a "first and last mile" option in areas currently underserviced by public transport; otherwise, a decrease in usage rates for public transport is expected [3,31]. Reduced participation in active transport is also commonly expected under this scenario [3,29,31] but could be prevented with strong investments in active transport infrastructure [3]. Overall, without strong policies and foresight on the part of governments, the interviewed stakeholders generally believed that AVs would result in unfavorable trends in terms of transport choices [3,29].

Survey studies have employed choice-based experiments that directly ask people which mode of transport they would use if AV options were available [20,32]. Most of these studies have found that people imagine they would prefer to use private AVs over shared AVs and public transport options [20,33–35], although one study found shared AVs to be preferred over private AVs [36]. Participants in the latter study were given a detailed breakdown of the potential costs of owning private vehicles versus shared AVs, which may have led the participants to consider the potential financial benefits of the shared format [36]. Limited survey research has included active transport as a transport option; one survey study examining whether people would use AVs instead of active transport revealed that 18% of participants thought they would use AVs for trips that they currently walk and 32% would use AVs for trips that they currently cycle [25]. One survey study that examined the willingness to use flying autonomous vehicles found that people might be willing to use them over other modes of transport if they were safe, fast, and easy to use [37].

The aim of the present study was to explore people's anticipated transport choices in a future where AVs are commonplace. To build on previous research, a novel approach was used, whereby respondents viewed a vignette describing a future scenario with autonomous vehicles, prior to deciding which mode(s) of transport they would use in this future scenario. It was expected that viewing the vignette would allow respondents to make more informed decisions about their envisaged transport choices in an autonomous future. Respondents were also provided with a wider range of transport choices compared to previous studies, a range that better reflects the options that are likely to be available.

## 2. Method

### 2.1. Sample

As part of a larger project examining the potential public health implications of autonomous vehicles (blinded for review), an ISO-accredited web-panel provider (Pureprofile) recruited a sample of 1078 Australian adults. Quotas were used to recruit a sample that was nationally representative in terms of age, gender, and location (metropolitan/regional residence). The sample profile is provided in Table 1. Potential respondents were emailed a generic invitation to participate in an online survey. Once they clicked the survey link, the respondents were advised that the survey examined how AVs might affect people's lifestyle behaviors. The completion rate among those commencing the survey was 77%. The study was approved by the university's human research ethics committee and all respondents provided informed consent.

**Table 1.** Sample profile (n = 1078).

| Attribute | n | % |
|---|---|---|
| Gender (female) | 541 | 50 |
| Male | 534 | 50 |
| Non-binary | 2 | <1% |
| Prefer not to say | 1 | <1% |
| Age | | |
| 18–35 | 333 | 31 |
| 36–54 | 339 | 31 |
| 55+ | 406 | 38 |
| Location (metropolitan) | 736 | 68 |
| Socioeconomic status ^ | | |
| Low (deciles 1–4) | 353 | 33 |
| Medium (deciles 5–7) | 337 | 31 |
| High (deciles 8–10) | 388 | 36 |
| Tertiary education | 724 | 67 |
| Personal vehicle user (at least weekly) # | 911 | 85 |
| Ride-share user (at least weekly) | 86 | 8 |
| Public transport user (at least weekly) | 292 | 27 |
| Any active transport participation (at least weekly) | 501 | 47 |
| Walking | 468 | 43 |
| Cycling/skateboard/scooter | 77 | 7 |
| E-bike or e-scooter | 31 | 3 |

^ Derived according to postcode, using the socioeconomic indexes for area classification (Australian Bureau of Statistics, 2018). # Car, truck/van, or motorcycle.

### 2.2. Survey Procedure

The respondents reported the following demographic information: age, gender, education, and postcode (to determine the location (metropolitan vs. non-metropolitan) and derive the socioeconomic status (SES) according to the Australian Bureau of Statistics [38]). Subsequent survey items assessed current travel behaviors, the perceived importance of transport-related factors, and attitudes to AVs. These variables have previously been related to transport choices [20,32]. Transport-related behaviors were examined by asking respondents which of the following types of transportation they would use in a typical

week: personal vehicle (car, truck, motorcycle, etc.), ride-hailing services (e.g., Ubers or taxis), public transport (e.g., bus, train, light rail, or ferry), micromobility (e.g., an e-bike or e-scooter), bicycle/skateboard/scooter, and walking. The importance of several factors in choosing the modes of transport was examined by asking "Please rate how important these factors are when you are choosing which mode of transport to use", with response options including cost, convenience, comfort, the inclusion of physical activity, safety, and environmental impact (1, "Very unimportant", to 5, "Very important"). To assess the respondents' attitudes to AVs, they were asked "How do you feel about fully autonomous vehicles being widely used in the future?" (1 "Very negative" to 5 "Very positive"). The full survey items are provided in the Supplementary Materials.

Given the probably limited familiarity with AVs presented by most respondents and their potential difficulty in imagining a future in which AVs are commonplace, the participants were then presented with a vignette depicting a potential future AV scenario. This was based on interviews conducted with 52 experts with professional backgrounds in transport, engineering, urban planning, and health. During the stakeholder interviews, they were asked to discuss how the introduction of AVs might affect numerous outcomes including health, transport, sustainability, and urban planning. For each stakeholder, the interview focused on their specific area of expertise. For example, government transport representatives were invited to discuss how AVs might affect road infrastructure, active transport infrastructure, public transport, urban design, and the usage rates for different modes of transport. The interviews were coded by the research team according to thematic analysis conventions [39] to identify the most common predictions for future AV scenarios, which subsequently informed the development of the vignette.

Vignette:

Imagine a world where people are no longer allowed to drive and all vehicles on public roads operate autonomously without drivers. These autonomous vehicles communicate with each other and with centralized computer systems, which allows them to operate swiftly and safely.

There are four autonomous vehicle options available to you:

1. You can own a personal autonomous vehicle, which is a highly convenient but expensive option.
2. You can use ride-hail autonomous vehicles (such as a self-driving taxi/Uber), which is a cheaper way to get around than owning your own autonomous vehicle and is also convenient.
3. You can use autonomous public transport options, which are very cheap and reliable. For example, autonomous shuttle buses are available that pick up/drop off people who are going in a similar direction.
4. You can buy or hire a personal automated flying vehicle, which is the most expensive option but is typically the fastest way to travel moderate distances.

Footpaths and cycleways are everywhere, making it easier to walk, cycle, and scoot to destinations. It is also very safe to travel this way because autonomous vehicles are highly effective at avoiding collisions.

This approach was consistent with previous research that has combined expert perspectives in this manner in order to envisage the role of AVs in the future [3,29,30,40,41]. After reading the vignette, the respondents were asked: "If you lived in this world, in a typical week, which of the following modes of transport would you use to get around?" They then selected which of the following transport options they would use: an autonomous personal vehicle, autonomous shared ride-hail vehicle, autonomous flying vehicle, autonomous public transport, micromobility (e-bike and e-scooter), bicycle/scooter/skateboard, and walking. Respondents could also select "None of these". Definitions of private and ride-hail AVs were provided as follows: "A private autonomous vehicle is a driverless vehicle that a person owns and uses for their own purposes" and "A ride-hail autonomous vehicle is a driverless vehicle that you don't own but can use, like an autonomous

taxi/Uber". A diagrammatic summary of the methodological approach is provided in the Supplementary Materials.

### 2.3. Analyses

Descriptive analyses were conducted to determine the proportion of respondents selecting each transport option at baseline and in an AV-based future. McNemar's tests were used to determine if the proportion of respondents selecting each option at baseline differed from the proportion selecting the equivalent option post-scenario exposure (except for "flying autonomous vehicle", which had no baseline equivalent). Binary logistic generalized linear models were used to identify the factors associated with using each transport option. The dependent variable for each model was the use of the respective transport option in the scenario (selected vs. not selected). The independent variables for each model were age (continuous), gender (male/female), location (metropolitan versus non-metropolitan), socioeconomic status (decile), AV-related attitudes (5-point scale), baseline transport behaviors ("use" versus the "do not use" each option), and the perceived importance of transport-related factors when choosing transport options (5-point scale).

## 3. Results

### 3.1. Transport Choices

The most frequently used transport options at baseline were personal vehicles (85% of respondents), walking (44%), and public transport (28%). For all future scenario transport options, the proportion of respondents who selected each option significantly differed from the baseline equivalent. Post-vignette exposure, the most popular options were walking (52%), an autonomous personal vehicle (41%), autonomous public transport (34%), and autonomous shared ride-hail AVs (24%). There were significant increases in the proportion of respondents selecting ride-hail options (7% to 24%), public transport (28% to 34%), active transport options (walking (43% to 52%) and bicycle/scooter/skateboard (7% to 14%)), and micromobility (3% to 14%). The proportion of respondents selecting a personal vehicle decreased (85% to 41%) (see Table 2).

**Table 2.** Baseline and anticipated use of transport options.

| Baseline | | Future Scenario | | |
| --- | --- | --- | --- | --- |
| Transport Option | % | Transport Option | % | *p*-Value |
| Conventional personal vehicle | 85 | Autonomous personal vehicle | 41 | <0.001 |
| Conventional ride-hail | 7 | Autonomous ride-hail | 24 | <0.001 |
| N/A | - | Autonomous flying vehicle | 10 | N/A |
| Conventional public transport | 28 | Autonomous public transport | 34 | 0.007 |
| Personal micromobility | 3 | Personal micromobility | 14 | <0.001 |
| Bicycle/scooter/skateboard | 7 | Bicycle/scooter/skateboard | 14 | <0.001 |
| Walking | 43 | Walking | 52 | <0.001 |

Note: *p*-values are derived from McNemar's tests comparing the proportion of respondents selecting each option at baseline and in the provided future scenario.

### 3.2. Predictors of Scenario Transport Choices

The logistic regression models revealed that the most consistent predictor of choosing a transport option in the future scenario was the current use of that option or the non-autonomous equivalent. The only exceptions were autonomous ride-hail services, which were typically selected by those who currently use a private vehicle, and autonomous flying vehicles, which had no baseline equivalent. Those who currently walk or use public transport were less likely to anticipate using private AVs in the scenario.

Those who had positive attitudes to autonomous vehicles being used in the future were more likely than other respondents to select the autonomous options. Respondents who valued convenience in transport were more likely to select autonomous public transport, whereas those who thought that the involvement of physical activity was an important

aspect of transport were less likely to choose public transport and shared ride-hail AV options. Older age was associated with a greater likelihood of using autonomous public transport and a reduced likelihood of using private AVs and bicycles/scooters/skateboards. Females were more likely than males to select autonomous shared ride-hail AVs, and respondents living in non-metropolitan areas were more likely to select walking, compared to those in metropolitan areas (see Table 3).

**Table 3.** Factors associated with the use of transport options in an autonomous scenario.

| Factors | Autonomous Personal Vehicle OR [95% CI] | Autonomous Ride-Hail OR [95% CI] | Autonomous Public Transport OR [95% CI] | Personal Micromobility OR [95% CI] | Bicycle/Scooter/ Skateboard OR [95% CI] | Walk OR [95% CI] | Autonomous Flying Vehicle OR [95% CI] |
|---|---|---|---|---|---|---|---|
| Age | 0.98 [0.97, 0.98] *** | 1.00 [1.00, 1.01] | 1.01 [1.00, 1.02] * | 1.00 [0.99, 1.01] | 0.98 [0.97, 0.99] *** | 1.00 [0.99, 1.01] | 0.99 [0.98, 1.01] |
| Gender (female) | 0.81 [0.62, 1.07] | 1.45 [1.07, 1.97] * | 1.14 [0.85, 1.53] | 1.35 [0.92, 1.97] | 0.79 [0.54, 1.17] | 1.15 [0.88, 1.51] | 0.66 [0.44, 1.01] |
| Regional location | 0.88 [0.64, 1.2] | 1.35 [0.96, 1.91] | 0.87 [0.62, 1.23] | 1.09 [0.7, 1.68] | 1.51 [0.97, 2.32] | 1.37 [1.01, 1.86] * | 0.87 [0.53, 1.43] |
| Socioeconomic status (decile) | 0.97 [0.93, 1.03] | 1.06 [0.99, 1.12] | 0.98 [0.92, 1.03] | 1.03 [0.96, 1.11] | 1.00 [0.93, 1.07] | 0.99 [0.94, 1.04] | 1.07 [0.99, 1.16] |
| Baseline use of | | | | | | | |
| Conventional car | 3.6 [2.28, 5.68] *** | 1.77 [1.11, 2.83] * | 1.01 [0.67, 1.51] | 1.72 [0.96, 3.1] | 0.92 [0.55, 1.54] | 0.88 [0.59, 1.32] | 1.1 [0.62, 1.96] |
| Conventional ride-hail | 1.09 [0.65, 1.84] | 1.55 [0.91, 2.62] | 0.90 [0.53, 1.52] | 1.23 [0.63, 2.4] | 1.31 [0.69, 2.47] | 0.63 [0.37, 1.07] | 2.02 [1.08, 3.78] * |
| Conventional public transport | 0.66 [0.46, 0.94] * | 1.33 [0.91, 1.94] | 4.3 [3.03, 6.1] *** | 0.93 [0.58, 1.51] | 0.94 [0.59, 1.52] | 1.83 [1.29, 2.59] *** | 1.01 [0.61, 1.69] |
| Micromobility | 1.04 [0.45, 2.4] | 1.17 [0.49, 2.81] | 1.23 [0.54, 2.83] | 15.64 [6.69, 36.56] *** | 1.15 [0.42, 3.14] | 0.81 [0.35, 1.88] | 1.13 [0.36, 3.49] |
| Bicycle/scooter/skateboard | 0.71 [0.41, 1.26] | 1.31 [0.74, 2.33] | 1.57 [0.90, 2.73] | 2.00 [1.06, 3.77] * | 10.84 [6.22, 18.86] *** | 1.53 [0.86, 2.73] | 1.39 [0.66, 2.89] |
| Walking | 0.64 [0.48, 0.86] ** | 1.16 [0.85, 1.60] | 1.72 [1.27, 2.33] *** | 1.37 [0.92, 2.05] | 1.95 [1.29, 2.94] ** | 4.38 [3.3, 5.81] *** | 0.64 [0.41, 1.01] |
| Positive attitude to AVs | 1.64 [1.43, 1.89] *** | 1.42 [1.23, 1.66] *** | 1.44 [1.25, 1.67] *** | 1.14 [0.95, 1.37] | 0.94 [0.78, 1.13] | 0.89 [0.78, 1.01] | 1.29 [1.05, 1.59] * |
| Importance of transport variables | | | | | | | |
| Cost | 1.03 [0.89, 1.19] | 1.12 [0.95, 1.32] | 1.15 [0.98, 1.35] | 1.05 [0.85, 1.29] | 1.16 [0.94, 1.44] | 1.00 [0.87, 1.16] | 0.97 [0.77, 1.22] |
| Convenience | 0.86 [0.72, 1.01] | 0.97 [0.80, 1.16] | 1.26 [1.05, 1.51] * | 0.91 [0.73, 1.14] | 0.97 [0.76, 1.23] | 1.13 [0.96, 1.33] | 0.86 [0.68, 1.1] |
| Comfort | 1.14 [0.96, 1.36] | 1.04 [0.86, 1.26] | 0.90 [0.74, 1.08] | 1.00 [0.78, 1.26] | 1.02 [0.8, 1.31] | 0.90 [0.76, 1.07] | 0.79 [0.62, 1.01] |
| Inclusion of physical activity | 1.03 [0.89, 1.2] | 0.81 [0.69, 0.96] * | 0.77 [0.66, 0.90] *** | 0.96 [0.78, 1.18] | 0.97 [0.78, 1.19] | 0.96 [0.83, 1.12] | 1.09 [0.87, 1.37] |
| Safety | 0.98 [0.84, 1.16] | 0.97 [0.81, 1.17] | 1.09 [0.91, 1.30] | 0.96 [0.77, 1.19] | 1.02 [0.81, 1.28] | 1.05 [0.89, 1.23] | 1.24 [0.97, 1.58] |
| Environmental impact | 0.94 [0.81, 1.1] | 0.93 [0.79, 1.10] | 1.00 [0.85, 1.17] | 0.96 [0.78, 1.18] | 0.93 [0.76, 1.13] | 1.00 [0.87, 1.16] | 0.95 [0.76, 1.19] |

Note: * $p < 0.05$, ** $p < 0.01$, *** $p < 0.001$.

## 4. Discussion

The results provide important insights into the anticipated travel choices in a future that incorporates autonomous vehicles. In line with the predictions made in previous research, private AVs were one of the most popular transport options [9,27]. However, the anticipated use of private AVs was less than half that of the current usage rate of conventional personal vehicles, suggesting that the use of private vehicles could decrease significantly once automated transport options are available. In comparison, the use of alternative transport options, including shared ride-hail AVs, active transport, and public transport, increased compared to current use. This contrasts with most previous modeling- [9,27], stakeholder- [3,29,30], and survey-based research [33–35]. These differences in results may, at least partly, be due to the scenario approach adopted in the present study, and further research is needed to assess the relative veracity of these differing data collection approaches.

The nature of the scenario description might explain why nearly one in four respondents envisaged using shared ride-hail AVs, despite the availability of private AVs. It was stated in the scenario that the shared ride-hail option would be cheaper than privately owned AVs, while still being convenient, which is consistent with predictions for these services in the future [42,43]. Research generally highlights cost as being an important determinant of people's transport choices [44–46]; a previous study highlighting the cost advantage of shared AVs also found that people were then more receptive to using them [36]. These results suggest that if shared AVs are cheaper than owning an AV but still provide a convenient service, this might prevent the overuse of private AVs. Policies that increase the relative cost of private compared to shared AVs (e.g., road charges) could encourage the greater adoption of shared AV options [47,48]. Implementing such policies should increase the sustainability of AVs and minimize the congestion that could be caused by the introduction of these new modes of transport [17,18].

The scenario presented to respondents also depicted a future with convenient active transport infrastructure and improved safety for vulnerable road users due to AVs. The respondents appear to have been receptive to this idea, with the envisaged use of all active transport options significantly increasing, compared to their current usage rates. Previous research has shown that developing an active transport infrastructure can increase participation in these forms of transport [49,50]. Therefore, in a future where AVs provide highly convenient door-to-door transport, making active modes of transport more feasible via infrastructure improvements is likely to be important in preventing population-level declines in physical activity [19,25,51]. Fostering physical activity in this way should help to reduce the incidence of non-communicable diseases, such as heart disease and stroke, while also being beneficial in terms of mental health outcomes [22–24].

More than four times as many respondents (14%) thought that they would use micromobility options such as e-bikes and e-scooters in the future described in the scenario, compared to their current use (3%). This could reflect the growing popularity of these vehicles, as has been observed over the last decade [52], and their compatibility with the highly connected active transport infrastructure described in the scenario. In the present study, older adults were less likely to use a non-electrified bicycle/scooter/skateboard, potentially reflecting the functional limitations that typically increase with age [53]. Micromobility options that use electric propulsion to supplement or replace physical activity can allow people who are less physically able to benefit from active transport infrastructure by improving their mobility, reducing isolation, and encouraging achievable levels of physical activity [54], thereby ensuring that more people can benefit from any improvements in active transport infrastructure. Government strategies that encourage the use of micromobility options, such as by improving active transport infrastructure and linking it to public transport options and popular destinations [55], could help to improve the mobility of people with physical limitations.

The observed increase in public transport patronage could partly be explained by the incorporation of shared autonomous buses into public transport systems in the given scenario. Stakeholders have predicted that if AVs are used to improve the accessibility and convenience of public transport systems, the use of these services is less likely to decline [3,31]. In line with this finding, respondents who rated convenience as an important factor in choosing transport options were more likely to select public transport after being exposed to the vignette. The respondents may have considered the inclusion of shared AVs to be a significant improvement to public transport and that this will increase accessibility and utility. These findings add further credibility to experts' recommendations for proactive government policies that leverage autonomous technologies to improve public transport, as a way to prevent these systems being superseded by AVs [3,29,31]. Improving usage rates for public transport in an autonomous future will help to increase the efficiency and sustainability of transport systems [10,19–21].

## 5. Limitations and Future Research

The primary limitation of the present study is that the validity of the findings is somewhat contingent on the accuracy of the vignette in representing the AV future. If the reality of AVs differs substantially from the described scenario, it is likely that people's transport behaviors would be different from those anticipated here. However, any predictions about transport behaviors in an as-yet-unrealized autonomous future are limited by these uncertainties [3,11], and the inclusion of experts across several disciplines during the development of the vignette is likely to have enhanced the validity of the depicted scenario. Second, the anticipated transport-related behavioral changes are hypothetical and may not translate into actual behavioral change, due to the intentions–behavior gap [56]. Third, the use of a web panel provider may have resulted in a sample that is skewed regarding unobserved characteristics (e.g., receptiveness to technology) and excludes those who do not have internet access. The sample was also restricted to Australian residents, meaning that the results might not pertain to different geographic and cultural contexts. Australia is

heavily reliant on private car ownership for transport and is characterized by sprawling low-density suburbs in metropolitan areas [57]. The results are going to be more applicable to those contexts that share similar features.

## 6. Conclusions

The present results suggest that rather than being detrimental, the advent of AVs could be beneficial for a range of outcomes. While this view is optimistic, if policymakers pursue a future that resembles the scenario presented to the respondents in this study, people may be more likely to increase their use of sustainable and healthy transport options that will improve efficiency in transportation systems. Public information programs that highlight the benefits of shared and active transport systems may help to realize this future [8,58]. The advent of AVs could be a turning point that can be leveraged to change transport behaviors for the better by creating environments that encourage favorable choices. However, such a future is unlikely to come to fruition organically, and if it is to be realized, this will require the proactive implementation of effective government policies that serve to encourage the use of alternate transport options over that of privately owned AVs.

**Supplementary Materials:** The following supporting information can be downloaded at: https://www.mdpi.com/article/10.3390/su151411245/s1.

**Author Contributions:** Conceptualization, L.B. and S.P.; methodology, L.B., S.P., J.T., J.B. and C.K.; formal analysis, L.B.; writing—original draft preparation, L.B.; writing—review and editing, S.P., V.F., J.T., R.V., B.G., J.B. and C.K.; project administration, L.B. and S.P.; funding acquisition, S.P., J.T., J.B. and C.K.; supervision, S.P. All authors have read and agreed to the published version of the manuscript.

**Funding:** This research was funded by the Australian National Health and Medical Research Council (grant number 2002905) and the Australian Research Council (grant number DP230102623). Jason Thompson receives salary support from the Australian Research Council (DECRA Fellowship DE180101411).

**Institutional Review Board Statement:** This study was conducted in accordance with the Declaration of Helsinki and was approved by the University of New South Wales Human Research Ethics Committee (approval number HC210207, 30 April 2021).

**Informed Consent Statement:** Informed consent was obtained from all subjects involved in the study.

**Data Availability Statement:** Due to the ethics committee's approval conditions, data are only available on request from the corresponding author.

**Conflicts of Interest:** The authors declare no conflict of interest.

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
