# Peer review of "Anticipated Transport Choices in a World Featuring Autonomous Transport Options"

_sustainability, doi:10.3390/su151411245_

Round 1

Reviewer 1 Report

The study provides valuable insights into the anticipated transport choices of individuals in an Australian context, considering the advent of autonomous vehicles. The methodology is appropriate, and the results shed light on potential shifts in transport behaviors in a future scenario with improved public transport, active transport infrastructure, and shared autonomous vehicles. However, there are a few points that need to be addressed before finalizing the publication.

Introduction and Background:

a) The introduction provides a general overview of the topic, but it would benefit from a more concise and focused introduction to the research problem and the specific research questions addressed in this study.

b) It would be useful to include a brief discussion of the existing literature on autonomous transport and its potential impact on mobility choices. This will provide better context for the study and highlight the novelty or contribution of the present research.

Methodology:

a) The description of the survey instrument used to collect data is missing. Please provide details on the survey questions related to demographic information, current transport behaviors, perceived importance of transport-related factors, and attitudes towards autonomous vehicles.

b) It is unclear how the vignette was developed and what specific elements were included to depict the future scenario with autonomous vehicles. Please provide more information on the vignette's content and how it was validated or reviewed by expert stakeholders.

c) The sample size and sampling method should be mentioned to ensure the study's representativeness and generalizability. If the sample is not representative, the limitations and potential biases should be acknowledged.

Results:

a) While the results provide a summary of the anticipated transport choices in the future scenario, it would be helpful to include more specific details and percentages regarding the changes in transport choices. This will enhance the clarity and comprehensiveness of the findings.

b) The regression models used to identify predictors of choices should be described in more detail. Specify the independent variables considered and the statistical methods used to assess their significance and predictive power.

Discussion and Conclusion:

a) The discussion section should provide a more thorough interpretation and analysis of the results. Explore potential reasons behind the observed changes in transport choices and consider the implications for health, sustainability, and transport system efficiency.

b) The conclusion should summarize the key findings and their implications. Discuss the policy implications of the results and highlight potential strategies for realizing the positive societal changes associated with autonomous transport.

Final Decision: Major Revisions Required

The study addresses an important knowledge gap regarding anticipated transport choices in the context of autonomous vehicles. However, several revisions are necessary to strengthen the manuscript. The introduction should be focused and provide a clearer research problem and objectives. The methodology needs to include details about the survey instrument and the development/validation of the vignette, as well as information on the sample size and sampling method. The results should provide more specific percentages and details, and the regression models should be described in more detail. The discussion and conclusion should be expanded to provide a more in-depth analysis of the results and their implications. Once these revisions are made, the manuscript will be reconsidered for acceptance.

Moderate editing of English language required

Reviewer 2 Report

I enjoyed reading this paper. It is an attempt to understand the preferences of people towards Avs in the future. Especially, in a scenario where automation has replaced conventional vehicles. It is indeed quite attractive! The paper is well-written. I have some comments though that might help.

First, I would like you to add a paragraph describing your paper structure.

Moving to the 2nd section, a diagram depicting your methodological steps, will be useful for the readers. What is more, what do you define as walking? Can you specify this? Did you consider a distance threshold? For instance, greater than 100m? Another question, how did you find this scenario? You mention something about 52 experts, but this should be further elaborated for clarity reasons. Is this scenario a product of a backcasting approach? Furthermore, why vignette is only text? Maybe an image will be more helpful. Is it due creativity reasons? Please justify this choice.

Moving to results, how do you explain this “respondents living in non-metropolitan areas were more likely to select walking compared to those in metropolitan areas”? It is quite a finding! Moreover, Table 3 is not fully presented. Please fix that. Also, some values are in bold font, but they don’t have the star of significance. Why? More specifically, factor conventional transport with Autonomous public transport 4.3 [3.03, 6.1]

You discussion is good; however, can you add a paragraph about potential policy suggestions to make scenario 1 a reality? Why did you decide to present “limitations” as a separate section.

In this section, there should be a comment about why you chose to present only one alternative scenario.

Furthermore, please add further research and enrich the contribution of your research! I have also another question, you mention that these findings might not pertain to other contexts. But maybe similar to Australia? Or not?

maybe to contexts similar to Australia? or not?

As for your last section, apart from government policies, it could be beneficial to mention community engagement as a means for implementing desirable future scenarios.

P.S. There are two papers that you can consult on future urban road.

Tsigdinos, S., Tzouras, P. G., Bakogiannis, E., Kepaptsoglou, K., & Nikitas, A. (2022). The future urban road: A systematic literature review-enhanced Q-method study with experts. Transportation Research Part D: Transport and Environment, 102, 103158. https://doi.org/10.1016/j.trd.2021.103158

González-González, E., Cordera, R., Stead, D., & Nogués, S. (2023). Envisioning the driverless city using backcasting and Q-methodology. Cities, 133, 104159. https://doi.org/10.1016/j.cities.2022.104159

Round 2

Reviewer 1 Report

 Accept in present form